# Coal Ash Enrichment with Its Full Use in Various Areas

**DOI:** 10.3390/ma15196610

**Published:** 2022-09-23

**Authors:** Victoria Petropavlovskaya, Tatiana Novichenkova, Mikhail Sulman, Kirill Petropavlovskii, Roman Fediuk, Mugahed Amran

**Affiliations:** 1Tver State Technical University, 170026 Tver, Russia; 2Polytechnic Institute, Far Eastern Federal University, 690922 Vladivostok, Russia; 3Peter the Great St. Petersburg Polytechnic University, 195251 St. Petersburg, Russia; 4Department of Civil Engineering, College of Engineering, Prince Sattam bin Abdulaziz University, Alkharj 11942, Saudi Arabia; 5Department of Civil Engineering, Faculty of Engineering and IT, Amran University, Amran 9677, Yemen

**Keywords:** recycling, energy, fuel, enrichment, separation, component, coal, ash

## Abstract

Increasing the percentage of recycling of various industrial waste is an important step towards caring for the environment. Coal ash is one of the most large-tonnage wastes, which is formed as a result of the operation of thermal power plants. The aim of this work is to develop a technology for the complex processing of coal ash. The tasks to achieve this aim are to develop a technology for the complex enrichment and separation of coal ash into components, with the possibility of their use in various applications, in particular: processing the aluminosilicate part as a pozzolanic additive to cement; carbon underburning for fuel briquettes; the iron-containing part for metallurgy and fertilizers. Complex enrichment and separation into components of coal ash were carried out according to the author’s technology, which includes six stages: disintegration, flotation, two-stage magnetic separation, grinding, and drying. The aluminosilicate component has a fairly constant granulometric composition with a mode of 13.56 μm, a specific surface area of 1597.2 m^2^/kg, and a bulk density of 900 kg/m^3^. The compressive strength for seven and twenty-eight daily samples when Portland cement is replaced by 15% with an aluminosilicate additive, increases to 30–35%. According to the developed technology, high-calorie fuel briquettes are obtained from underburnt with a density of 1000–1200 kg/m^3^, a calorific value of 19.5–20 MJ/kg, and an ash content of 0.5–1.5%. The iron-containing component, recovered by two-stage magnetic separation, has the potential to be used in metallurgy as a coking additive, in particular for the production of iron and steel. In addition, an effective micro-fertilizer was obtained from the iron-containing component, which: is an excellent source of minerals; improves the quality of acidic soil; helps soil microorganisms decompose organic matter faster, turning it into elements available to plants; promotes rooting of seedlings; helps to more effectively deal with many pests and diseases. As a result, the complete utilization of coal ash in various applications has been achieved.

## 1. Introduction

Solving the problem of recycling various production waste fully meets the priority sustainable development goals [1]. This will preserve the natural environment, improve the ecological situation, reduce carbon dioxide emissions, and reduce the area occupied by waste [2]. As a result of the activities of enterprises in the fuel and energy complex, the most significant coal ash is formed [3]. Such waste occupies vast areas, contributes to environmental pollution, and can be the cause of many serious natural disasters and pollution of air and water sources [4]. At the same time, they can be successfully used in the construction industry, which constantly needs large amounts of raw materials [5]. Coal ash is used for a variety of purposes, such as cement additive [6], road base [7,8], soil additive [9], zeolite synthesis [10], raw material for the extraction of rare earth elements [11], and as an absorbent [12].

In the world literature of recent years, the results of studies on the use of industrial waste in various proportions in the manufacture of binders for concrete are presented in sufficient detail [13,14,15]. Articles [16,17,18] developed a wide range of binders with low water demand using polymineral raw materials. Metakaolin [19], microsilica [20], fly ash [21], etc., have been sufficiently studied in this aspect. On the other hand, throughout the country, millions of tons of coal ash are stored in large quantities at ash dumps, occupying tens and hundreds of hectares of land [22]. Due to the low activity and little research on these wastes, only about 10% of coal ash is reused (Figure 1).

As part of ensuring the use of “rational models of consumption and production” in the field of building materials science, it is necessary to comprehensively develop the possibility of using production waste to replace natural raw materials in the production of building products [26]. The low activity of coal ash is due to the fact that it has been in dumps for many years and is exposed to atmospheric precipitation [27]. However, the consideration of coal ash as waste is not rational and does not correspond to modern trends in environmental protection and building materials science [28]. Known technologies for the use of waste fuel ash and slag to replace parts of the cement or obtain without-cement compositions [29,30]. Cement production is a CO_2_-intensive process due to the natural formation of carbon dioxide during the firing process by the decomposition of limestone, which is additional to the CO_2_ produced by burning the fuel [31]. In view of the fact that, at present, one of the main problems in the production of Portland cement is the control of carbon dioxide emissions, the involvement of waste ash as part of the binder can reduce this negative impact [32]. It is known that the production of one ton of cement contributes to the formation of 918 kg of CO_2_, which leads to the greenhouse effect and global warming [33,34].

The involvement of the maximum possible amount of ash and slag in the production of such a highly demanded binder as cement could not only improve the environmental situation and reduce CO_2_ emissions [35]; the introduction of fuel waste into the composition of cement contributes to the improvement of their operational properties such as sulfate resistance, reduction in heat generation, etc. [36]. In the modern world, this task is relevant and in demand [37,38].

The structuring role of fuel ash in building compositions, according to many authors, is due to the type and technology of fuel combustion, as well as the method of removing ash waste [39]. However, the condition of the ash or slag does not always meet the specifications for fillers and may require additional enrichment or the introduction of auxiliary operations [40]. They increase the efficiency of waste ash utilization [41]. Researchers often use milling to increase the activity of fly ash [42]. Thus, the joint grinding of cement binder, fly ash, limestone, and chemical additives is reflected in the intensification of hydration processes during hardening and increases the activity of the composite binder by up to 62%, as studies show [43]. The strength of cement with the addition of ash at the age of 3 days of natural hardening can be two times higher than the strength of the control composition [44]. However, ash milling technology should be treated with caution as it is not always a sustainable approach. Especially given the current energy challenges, each of these operations add significant energy costs. When designing compositions, researchers should (preliminarily) evaluate the feasibility and economic feasibility of their choice, not only from a research point of view but also from an industrial and applied point of view. From this, similar assessments should be made for other proposed operations, thereby confirming the real effectiveness of the proposed technology. On the other hand, without grinding or other enrichment, it is impossible to turn waste into secondary resources.

Researchers [45] also note a change in the rheological properties of the cement mixture; however, these data like many others [46,47] refer to the most demanded and studied waste such as fly ash with a high calcium content. Acid ash with low calcium content is less applicable [48]. These wastes in modern construction technologies are not yet widely used [49]. Ashes from hydro removal and now during storage occupy huge areas of the urban environment [50].

Since according to the technology such ash waste is removed using water, this affects its properties [51]. Such ash is characterized by a heterogeneous composition, low calcium content, and the presence of a sufficiently large volume of impurities [52]. The issue of the ash composition variability is raised before researchers, due to the fact that this can affect the effectiveness of mechanical processing methods. For enrichment technology to be reliable and flexible, it must be complex and involve more than just grinding. This causes difficulties in the processing of acidic fly ash for the production of modern cement and concretes, from the variable compositions of particles in size [53]. First of all, this concerns the content of particles in the range of the largest fractions in the composition of the ash and slag mixture [54]; therefore, the ash waste from hydraulic removal must be separated into separate fractions. Thus, it will be possible to use each individual component exactly where it will make the greatest contribution to the creation of building products [55]. Separation, flotation, dispersion, activation in various grinding devices, and other methods of ash processing contribute to its efficiency [56].

Of course, direct and immediate use of bulk ash is possible for the production of various building materials, for example in road construction (e.g., asphalt composition). It is more economical to prioritize the direct use of waste, avoiding additional costs, especially for low-value end materials. However, given the potential activity of enriched coal ash, it is unreasonable to “bury” this valuable resource, for example, as bedding under road pavement. Moreover, the huge areas of ash dumps force us to look for and use all possible methods of disposal.

It should be noted that the characteristics of ash and slag mixtures will vary in a fairly wide range depending on the type of fuel used, the technology of dispersion of solid fuel before loading into the furnace, the technological process (primarily temperature and time) of combustion, the method of removing ash and slag waste, etc. [57]. In fact, the ash and slag mixture is mixed in different proportions of fly ash and fuel slag [58]. The formula for determining the quality of coal ash depends on the chemical composition (1):M_b_ = (CaO + MgO)/(SiO_2_ + Al_2_O_3_)(1)
where M_b_ is the basic module according to which coal ash is classified. M_b_ > 1—basic, M_b_ < 1—acidic, M_b_ ≈ 1—neutral (Table 1).

The formula for determining the quality factor (2):K = (CaO + MgO + Al_2_O_3_)/(SiO_2_ + MnO)(2)

As follows from [59], the removal of excess iron oxide increases the activity of ash and slag materials. In particular, the use of fly ash in concrete increases the strength and the standard level of durability compared to traditional compositions [60]. However, as a result of hydraulic removal, ash and slag wastes are accumulated in ash dumps, and only 10% of fly ash is captured [61]. It is obvious that the problem of utilization of huge accumulations of ash and slag mixtures at ash dumps is ripe [62]. In the course of the analysis of the world literature, only technologies for the use of coal ash as a partial replacement of soil in the construction of road foundations were identified, which, is not able to effectively solve the problem of recycling this type of waste [63].

It is also known that coal ash is an iron-aluminosilicate mineral raw material, the composition of which is due to an amorphous matrix with inclusions in the form of microspheres [64]. Other advantages of coal ash compared to fly ash include a smaller amount of harmful impurities, due to the fact that they are gradually washed away by precipitation, in particular, acid rain [65].

As mentioned above, to date, enough scientific work has been accumulated aimed at using fly ash as a component of a composite binder but there is a lack of research on the use of hydro-removed coal ash in the production of building materials. Summing up, it is noted that the use of coal ash in polymineral composite binders has potential but is not well researched.

One of the most promising technologies is the regeneration of dump coal ash by drying with the capture of a fine fraction and its further grinding [66]. At the same time, the problem statement lies in the fact that complex and energy-consuming technologies of mechanical activation and enrichment minimize the effect of the use of coal ash [67]. Thus, new coal ash enrichment technologies are needed that can separate waste into various components suitable for use in various industries.

The aim of this work is to develop a technology for the complex processing of coal ash. The tasks to achieve this aim are to develop a technology for complex enrichment and separation of coal ash into components with the possibility of their use in various industries, in particular: the aluminosilicate part as a pozzolanic additive to cement; carbon underburning for fuel briquettes; the iron-containing part in metallurgy and agriculture.

This manuscript’s remaining sections are structured as follows. Section 2 details the materials and methods. In Section 3.1, a technology of coal ash complex enrichment and separation of components is shown. Section 3.2 is about the use of the aluminosilicate part as a pozzolanic additive to cement. Section 3.3 studied carbon underburning for fuel briquettes. Section 3.4 detailed the iron-containing part for metallurgy and agriculture. Section 4 summarized some conclusions from conducted research.

## 2. Materials and Methods

### 2.1. Materials

The object of the study is the coal ash from the Kashirskaya thermal power plant (Kashira, Russia) (Figure 2). The annual amount of coal ash emitted from this thermal power plant is 9 million tons. It was formed as a result of the combustion of brown coal and has parts of its mineral and organic components. Chemical analysis of the initial coal ash showed that the main components in it are oxides of silicon, aluminum, iron, and carbon (Figure 3). This trend persists for particles of all sizes. The particle size distribution is shown in Figure 4. It was established that most of the particles of coal ash have a size of less than 10 µm.

Portland cement CEM I 42.5 N (LafargeHolcim, Kolomna, Russia) was used as the initial binder. The chemical and mineral composition of the Portland cement is shown in Figure 5.

To maintain equal flowability of the mixtures with the introduction of the aluminosilicate component, the polycarboxylate superplasticizer Master Glenium 115 (BASF, Ludwigshafen, Germany) was used.

### 2.2. Mix Design

Figure 6 gives details on ash sampling, to ensure the representativeness of the samples.

Complex enrichment and separation into components of coal ash were carried out according to the author’s technology, which is described in detail in Section 3.1.

Grinding of the obtained aluminosilicates component was carried out by an Activator-4M planetary mill (Activator, Chelyabinsk, Russia) for 30 min. The impact of the ferromagnetic grinding media (Figure 7a) allows for efficient milling and activation of the powder (Figure 7b). Processing must be sustainable and economically viable to be of real industrial and practical interest. Against the background of other options for increasing the stability of this step, the use of a planetary mill allows, in the shortest possible time and at minimal cost, to grind and activate the powder due to the combined action of impact, centrifugal, and abrasive forces.

The effectiveness of the obtained aluminosilicate component of enriched coal ash was tested during its use as a replacement for cement in various proportions (Figure 8).

### 2.3. Methods

The complex enrichment and separation into components of coal ash were carried out according to the author’s technology, which is described in detail in Section 3.1.

Mathematical planning was carried out using the Develve 4.13.0.0 software package (Develve, Velp, The Netherlands). The representativeness of the samples and the set of necessary tests were carried out from modern positions; in this case, the error for all experiments was no more than 5%.

To study the elemental and mineral composition, a D8 Advance Bruker AXS X-ray powder diffractometer (wavelength λ = 1.5418 Å) was used. Granulometry of particles of raw materials was carried out using a laser analyzer, Analysette 22 (Fritsch, Idar-Oberstein, Germany). The specific surface of bulk raw materials was studied using the PSH-11 device (Khodakov Devices, Moscow, Russia).

Morphological features of the microstructure were studied using a Tescan MIRA3 scanning electron microscope (Brno, Czech Republic), which makes it possible to carry out energy dispersive spectroscopy.

The compressive strength was determined according to the Russian standard GOST 310.4-81 on cubes with an edge of 70 mm.

The scatter of the results obtained for all tests within each series of samples did not exceed 5%

## 3. Results and Discussion

### 3.1. Technology of Coal Ash Complex Enrichment and Separation of Components

Complex enrichment and separation of components of coal ash were carried out according to the author’s technology, which includes six stages: disintegration, flotation, two-stage magnetic separation, grinding, and drying (Figure 9).

In the first block, disintegration is accompanied by separation into fractions above and below 50 µm. The coarse fraction is sent to the dump, and the fine particles enter the flotation unit, where the underburnt is removed (Figure 10).

The resulting underburnt is briquetted and used as fuel (Figure 11). Table 2 lists the chemical composition of the underburnt.

Iron-containing particles are removed from the mixture by two-stage magnetic separation (Figure 12).

At the same time, in the first stage, the separator acts on the coal ash with a magnetic induction of 600 T, removing various contaminants together with iron-containing particles; at the next stage, a more precise cleaning with a magnetic induction of 400 T takes place.

Table 3 lists the chemical composition of the iron-containing component after each magnetic separation stage.

The non-magnetic fraction containing Al and Si oxides (Figure 12) accumulates and dehydrates. Under laboratory conditions, the dehydrated aluminosilicate mixture is dried in a muffle furnace at 300 °C for 1 h.

### 3.2. Use of the Aluminosilicate Part as a Pozzolanic Additive to Cement

The chemical composition of the obtained aluminosilicates is shown in Figure 13. The aluminosilicate component has a fairly constant particle size distribution with a mode of 13.56 μm (Figure 14). The appearance of the aluminosilicate component is shown in Figure 15.

Microscopic (Figure 16a) and X-ray fluorescence (Figure 16b) studies of aluminosilicate particles show that the structure of the resulting product is represented by a glass phase, in which some areas of crystalline formations of quartz are visible.

The specific surface of the enriched aluminosilicate product after grinding by an Activator-4M planetary mill is 1597.2 m^2^/kg and the bulk density is 900 kg/m^3^. Enrichment of the ash product provides an optimized dispersed system of the mineral binder with a large number of contacts. The aluminosilicate component of coal ash holds fairly stable chemical, physical, and mechanical properties. The place of the obtained aluminosilicate component in the “CaO-SiO_2_-Al_2_O_3_” system is shown in Figure 17.

One can see an increase in the amorphism of the aluminosilicate component compared to the initial coal ash (Figure 18), which will positively affect the reactivity.

The developed aluminosilicate component, in terms of its granulometry, chemical, and mineral composition has significant potential for use as an active additive in a composite binder. In particular, the compressive strength for 7- and 28-day samples, when Portland cement is replaced by 15% with an aluminosilicate additive, increases to 30–35% (Figure 19).

The resulting improvement in physical and mechanical properties from the inclusion of an aluminosilicate component in the composition allows us to more effectively control the properties of cement paste due to the active interaction of Portland cement minerals with aluminosilicate components of the coal ash. The inclusion of ash grains in the dispersed system of the binder makes it possible to regulate the internal structure of the paste. The studied mineralogical composition of the developed binder is shown in Figure 20. It is shown that the main components of the hardened ash-cement composition, after hardening, are quartz, microcline, portlandite, calcite, alite, belite, and tetracalcium alumoferrite; the remaining mineral components are contained in small amounts, less than 1%. The amorphous phase in the composition of the hardened paste is about 10%.

Thus, studies have established that the introduction of an ash aluminosilicate component into the composition of the cement affects the strength and density of the cement paste. The optimal content of the activated ash component (15%) has a positive effect on the structure of the resulting material. In the studied compositions, the chemical activity of ash is manifested. In the compositions, calcium hydroxide Ca(OH)_2_ binds in a larger volume into insoluble compounds. The physical effect of the aluminosilicate part of activated coal ash on the dispersed system of the mineral binder is also manifested. This approach contributes to obtaining the necessary packing of particles. The structure of the paste is compacted and strengthened due to the physical and chemical interactions of all components of the activated ash-cement compositions.

### 3.3. Carbon Underburning for Fuel Briquettes

According to the developed technology, high-calorie fuel briquettes are obtained from underburnt. These fuel briquettes are a good alternative to conventional wood or coal. The technology for the production of fuel briquettes is based on the process of pressing waste under high pressure (with or without heating). The production process consists of grinding raw materials, drying, and pressing. The resulting fuel briquettes do not include any binders, except for one natural such hydrolytic lignin (7 wt.%), contained in the cells of plant waste. Lignin is released at high pressure and heating and gives strength to the briquettes. The temperature present during pressing contributes to the melting of the surface of the briquettes, which due to this becomes more durable, and is important for the transportation of the briquette. One of the most popular methods for producing fuel briquettes is extrusion using special extruders.

The density of obtained fuel briquettes is 1000–1200 kg/m^3^, the calorific value is 19.5–20 MJ/kg, and ash content is 0.5–1.5%. Fuel briquettes are used as a solid fuel for fireplaces and stoves of all kinds, including solid fuel boilers for heating systems. Since fuel briquettes are an environmentally friendly product and burn almost smokelessly, it is ideal to use them for heating residential premises, baths, tents, greenhouses, vegetable pits, etc.

Briquetting is a good alternative to the direct use of straw and wood waste as fuel. Briquettes emit more heat than straw, sawdust, and wood chips in their pure form, increasing the efficiency of boiler houses; they do not require large storage areas and do not spontaneously ignite during storage. For example, when burning 1 ton of wood pellets, the same amount of energy is released as when burning 1.6 tons of wood, 480 m^3^ of gas, 500 L of diesel fuel, or 700 L of fuel oil.

### 3.4. Iron-Containing Part for Metallurgy and Agriculture

The iron-bearing component recovered by two-stage magnetic separation has the potential to be used in metallurgy as a coking additive, in particular for the production of iron and steel. In metallurgy, the use of this additive in metal smelting has a key role; this bulk additive allows a balance of the chemical composition of the metal and brings it to the required quality standards, while the following positive effects are noted:-Does not have a negative impact on the anti-corrosion properties of metals because the percentage of sulfur in them is minimal and they also retain strength and other characteristics of metals;-Does not contain nitrogen, which leads to the destruction of metals and the formation of cracks.

Not only iron-containing components are used in metallurgy, but production waste is also used in the aluminum industry [68].

The obtained iron-containing component is also an effective micro-fertilizer. This is due to the fact that this material is an excellent source of minerals, which: improves the quality of acidic soil; helps soil microorganisms decompose organic matter faster, turning it into elements available to plants; promotes the rooting of seedlings; helps to more effectively deal with many pests and diseases.

Due to the polymineral composition of the iron-containing component, the following positive effects are provided:-Improves metabolic processes, promotes more lush flowering;-Helps the plant to absorb vitamins; this compound is especially useful for bulbous plants because with its deficiency the bulbs begin to exfoliate and dry out;-Improves the resistance of plants to diseases and adverse climatic conditions; promotes the development and growth of roots;-Participates in the process of photosynthesis, promotes the formation of enzymes, increases the immunity of plants, and their frost resistance; improves soil uniformity;-Helps plants regulate water balance, and resist winter frosts; with an insufficient amount of this substance, ammonia accumulates in the leaves and roots, which slows down the growth of the plant;-Participates in the formation of carbohydrates, from which starch and cellulose are subsequently formed; contributes to the normalization of the water balance of plants, as well as the activation of enzymes.

This confirms the study [69] that provides a green engineering approach to recycling coal ash for regreening mines, as well as a new development direction for the high-value green recyclable pathway of coal ash.

In fact, it is possible to use the iron fraction obtained from a single magnetic separation step to develop low-cost agronomic applications. However, for metallurgical applications, a two-stage magnetic separation will be necessary because the quality of the final product must be high and well defined; therefore, without the inclusion of undesirable impurities. The chemical composition of the iron-containing component obtained after each stage, shown in Table 3, will help readers determine the required application of the materials.

## 4. Conclusions

At the moment, the problem statement lies in the fact that complex and energy-consuming technologies of mechanical activation and enrichment minimize the effect of the use of coal ash. The current work is aimed at the comprehensive disposal of large-tonnage coal ash waste generated as a result of the operation of thermal power plants, which fully meets the priority goals of sustainable development. The following main conclusions, according to the results obtained, were presented:A technology has been developed for complex enrichment and separation of coal ash into components with the possibility of their use in various industries, in particular: the aluminosilicate part as a pozzolanic cement additive; carbon underburning for fuel briquettes; the iron-containing part for fertilizers. Complex enrichment and separation into components of coal ash were carried out according to the author’s technology, which includes six stages: disintegration, flotation, two-stage magnetic separation, grinding, and drying;The aluminosilicate component has a fairly constant particle size distribution with a mode of 13.56 μm, a specific surface area of 1597.2 m^2^/kg, and a bulk density of 900 kg/m^3^. The compressive strength for seven and twenty-eight daily samples when Portland cement is replaced by 15% with an aluminosilicate additive increases to 30–35%;According to the developed technology, high-calorie fuel briquettes are obtained from underburnt with a density of 1000–1200 kg/m^3^, a calorific value of 19.5–20 MJ/kg, and an ash content of 0.5–1.5%. The resulting fuel briquettes do not include any binders, except for one natural such hydrolytic lignin (7 wt.%), contained in the cells of plant waste;The iron-containing component is an effective micro-fertilizer. This is due to the fact that this material: is an excellent source of minerals; improves the quality of acidic soil; helps soil microorganisms decompose organic matter faster, turning it into elements available to plants; promotes the rooting of seedlings; helps to more effectively deal with many pests and diseases. The iron-bearing component recovered by two-stage magnetic separation has the potential to be used in metallurgy as a coking additive, in particular for the production of iron and steel.As a result, the complete utilization of coal ash in various industries has been achieved.

## Figures and Tables

**Figure 1 materials-15-06610-f001:**
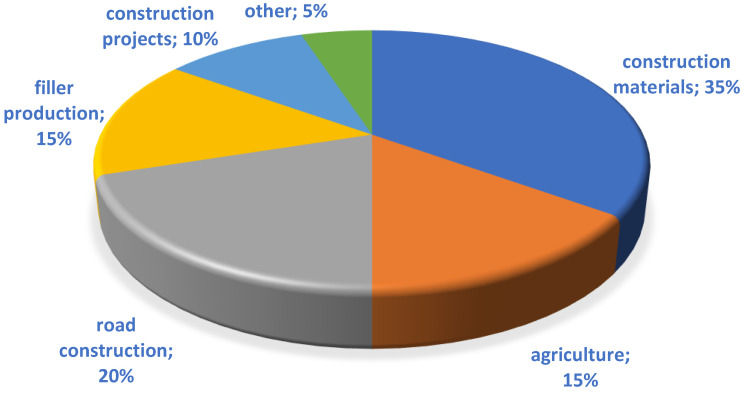
Applications of coal ash [23,24,25].

**Figure 2 materials-15-06610-f002:**
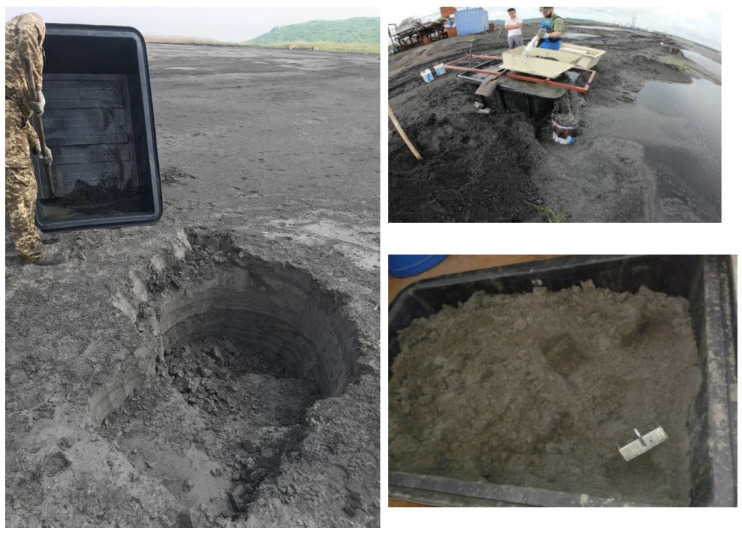
Selection of coal ash for research.

**Figure 3 materials-15-06610-f003:**
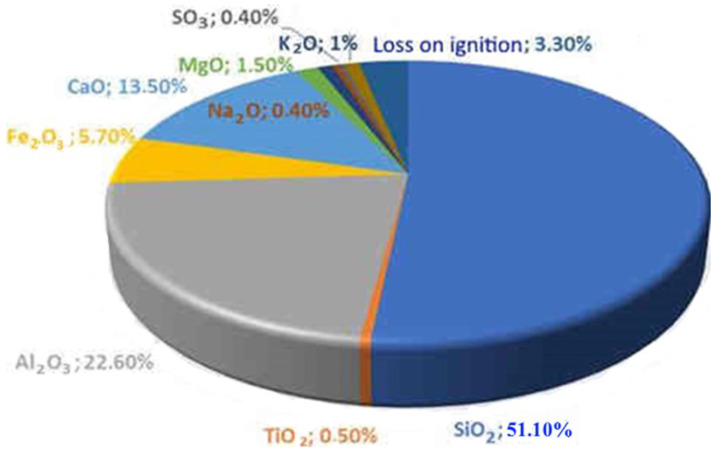
Chemical composition of the original coal ash.

**Figure 4 materials-15-06610-f004:**
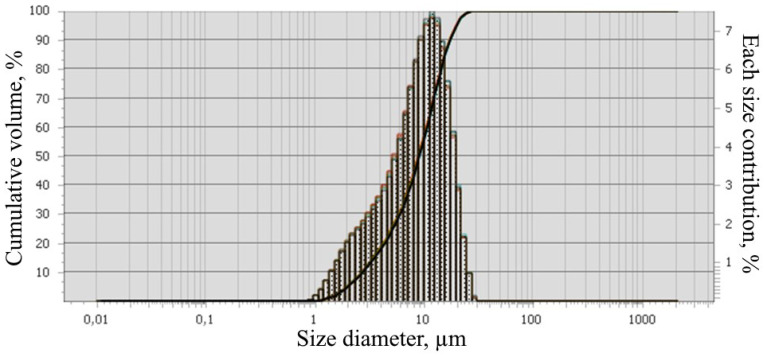
Particle size distributions of the initial coal ash.

**Figure 5 materials-15-06610-f005:**
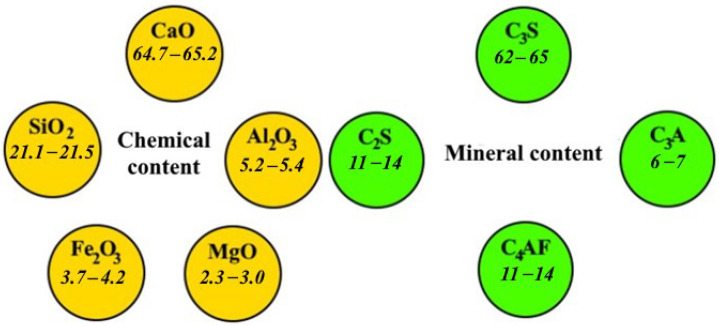
Chemical and mineral composition of the Portland cement, wt.%.

**Figure 6 materials-15-06610-f006:**
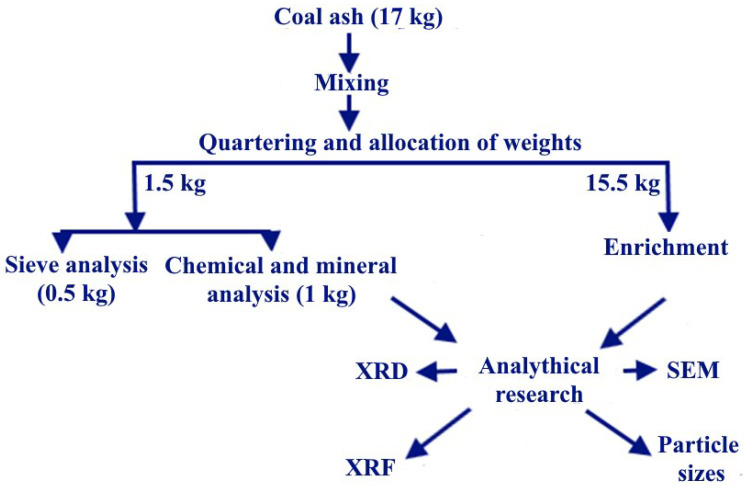
Details on ash sampling.

**Figure 7 materials-15-06610-f007:**
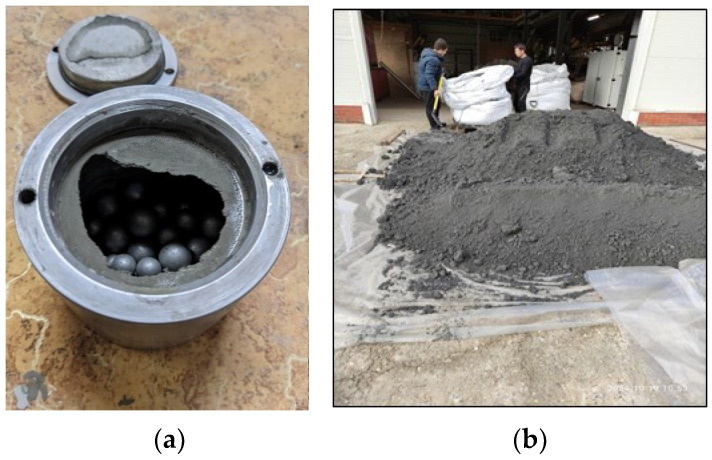
Ferromagnetic grinding media (**a**) and resulting powder (**b**).

**Figure 8 materials-15-06610-f008:**
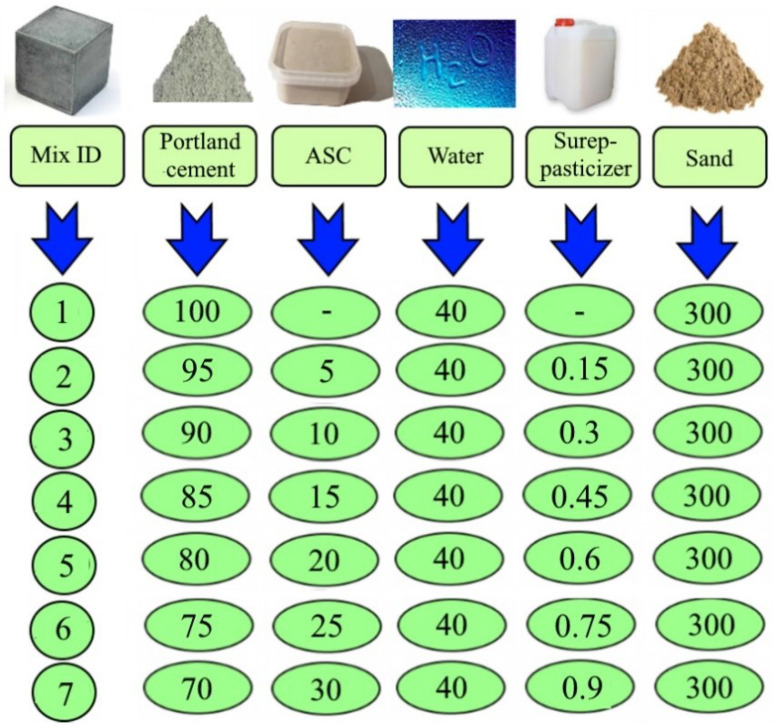
Mix proportions (in fractions).

**Figure 9 materials-15-06610-f009:**
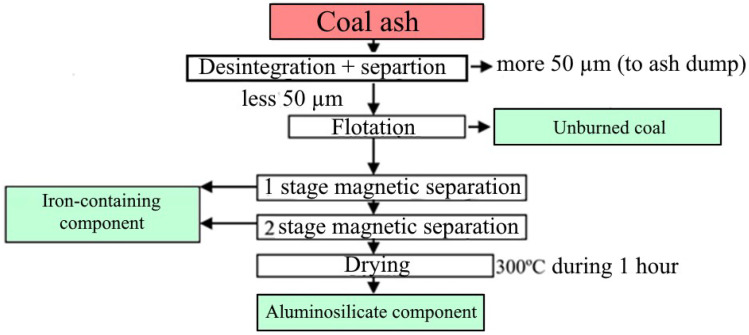
Developed technology of coal ash complex enrichment.

**Figure 10 materials-15-06610-f010:**
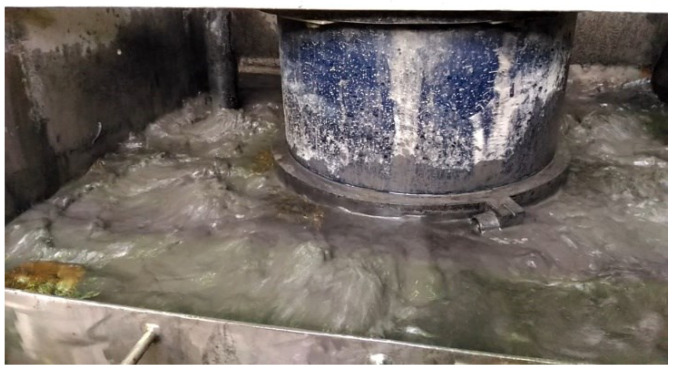
Removal of underburnt by flotation.

**Figure 11 materials-15-06610-f011:**
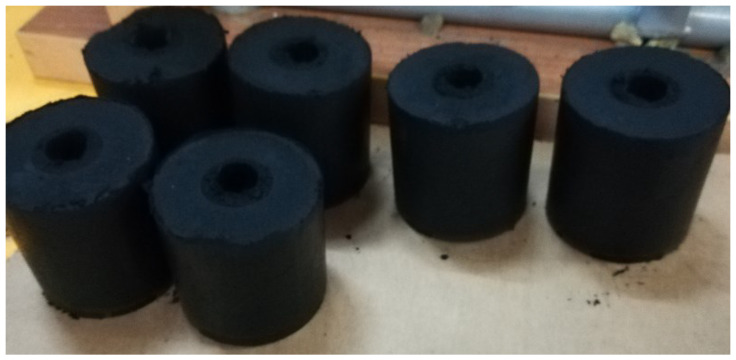
Briquetted underburnt.

**Figure 12 materials-15-06610-f012:**
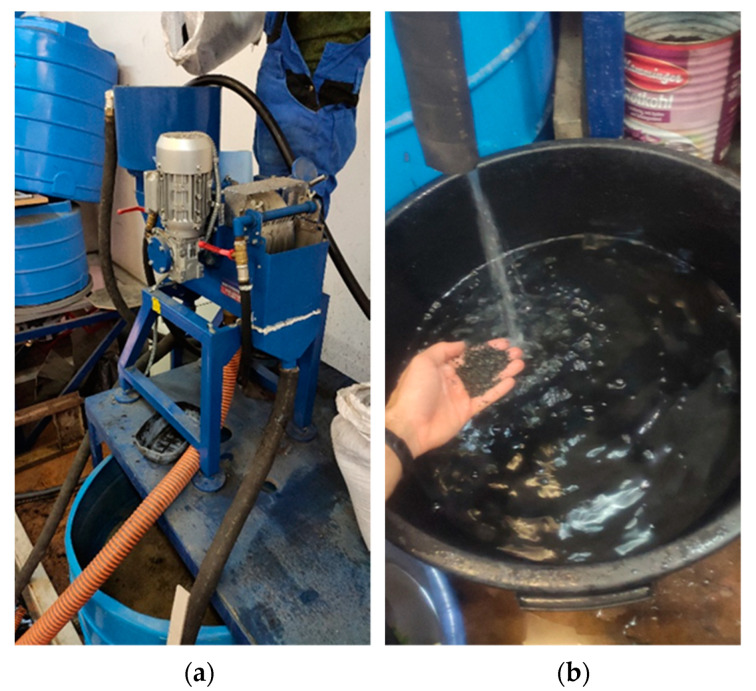
Magnetic separator (**a**), and obtained material (**b**).

**Figure 13 materials-15-06610-f013:**
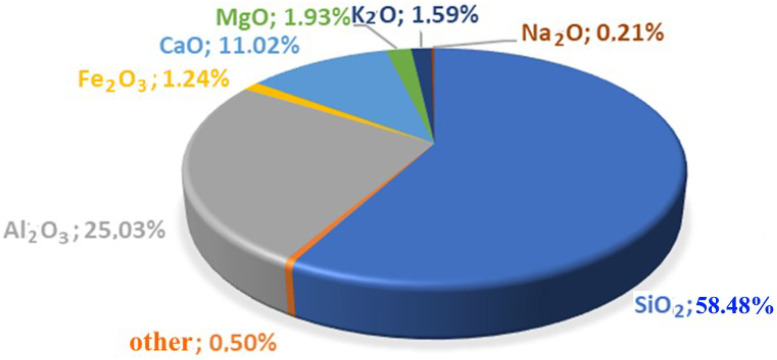
Chemical composition of the obtained aluminosilicates.

**Figure 14 materials-15-06610-f014:**
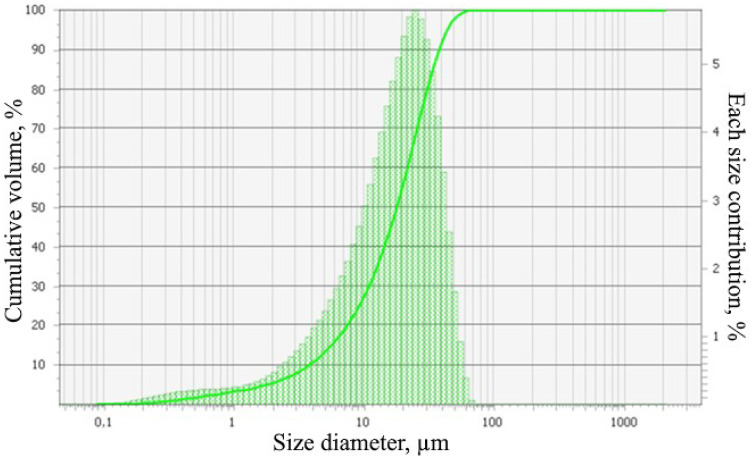
Particle size distributions of the aluminosilicate component.

**Figure 15 materials-15-06610-f015:**
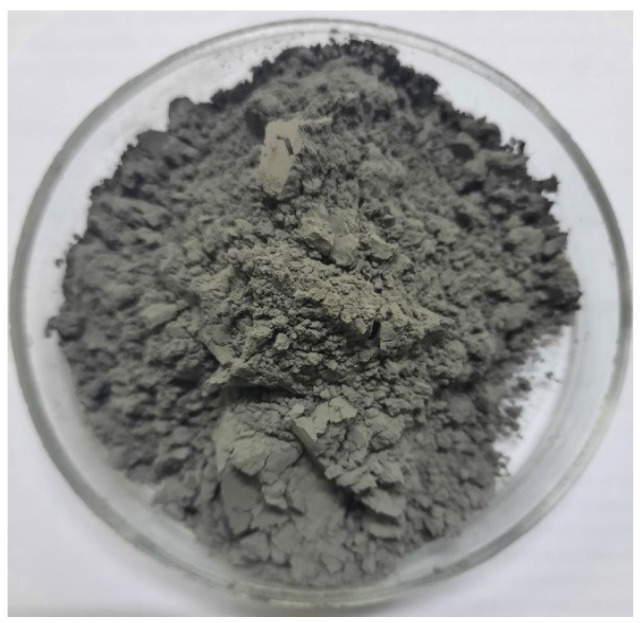
Appearance of the aluminosilicate component.

**Figure 16 materials-15-06610-f016:**
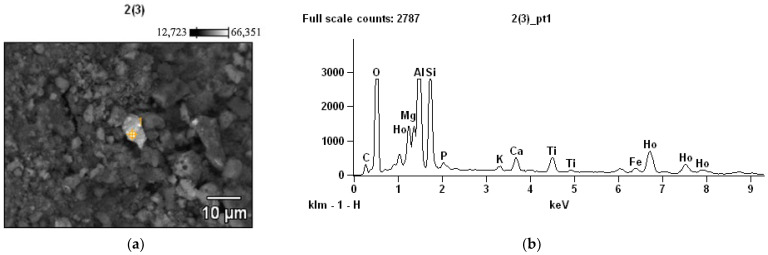
Microscopic: (**a**) and (**b**) X-ray fluorescence results of aluminosilicate particles.

**Figure 17 materials-15-06610-f017:**
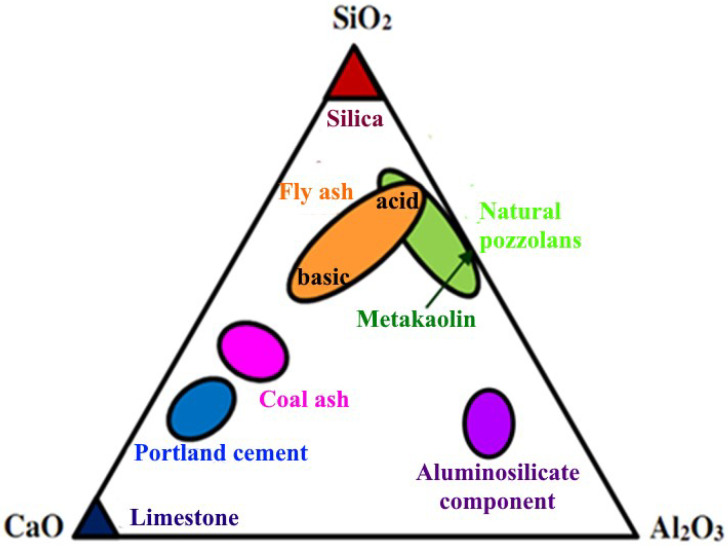
Place of the obtained aluminosilicate component in the “CaO-SiO_2_-Al_2_O_3_” system.

**Figure 18 materials-15-06610-f018:**
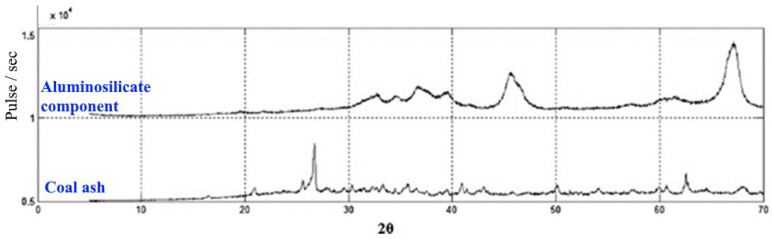
Comparison of XRD spectra of initial coal ash and aluminosilicate component.

**Figure 19 materials-15-06610-f019:**
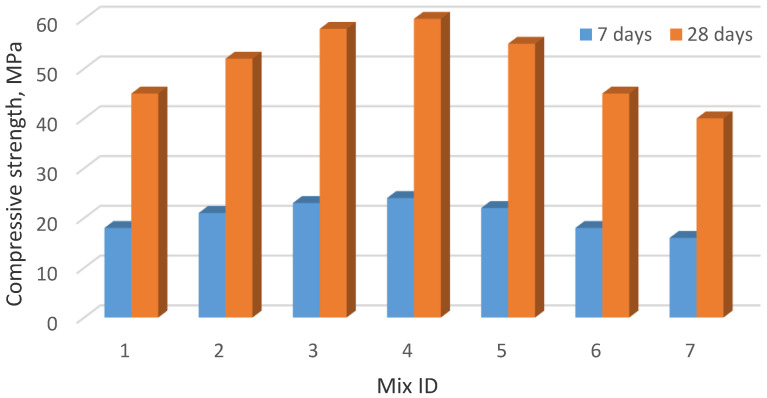
Change in the strength of the developed composites.

**Figure 20 materials-15-06610-f020:**
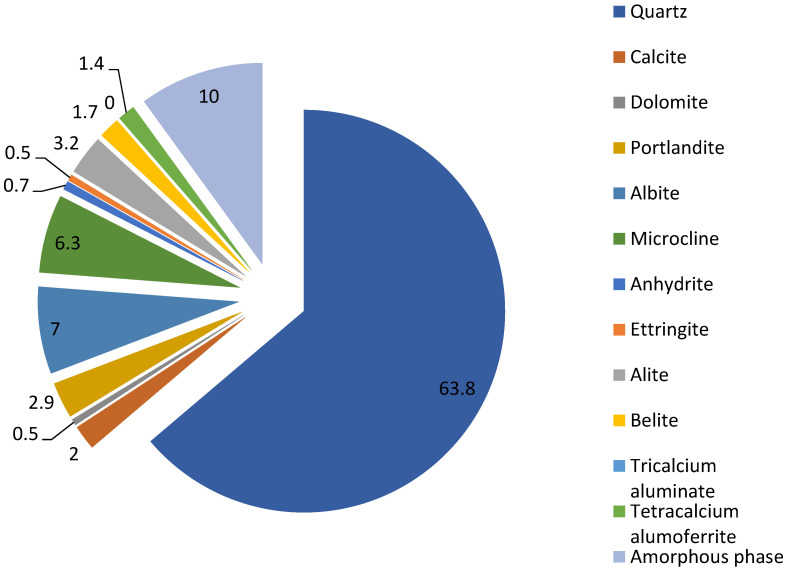
Mineral composition of the developed composite, wt.%.

**Table 1 materials-15-06610-t001:** Classification of coal ash by activity groups.

Characteristics		Groups of Ash and Slag Materials		
Chemical activity		active (high calcium)	covertly active	inert (low calcium)
Quality indicators	M_b_	>0, 5–2, 8	>0, 1–0, 5	<0, 1
	K	1, 0–3, 6	0, 5–1, 5	0, 4–0, 9
Vitreous phase color		brown and dark	any color	colorless
Activity		self-hardening	requires intensification of hardening	inert

**Table 2 materials-15-06610-t002:** Chemical composition of the underburnt.

Element	C	H	N	O	S
Content, wt.%	92.0	1.7	2.3	1.6	2.4

**Table 3 materials-15-06610-t003:** Chemical composition of the iron-containing component after each magnetic separation stage.

Stage	Fe_2_O_3_	Al_2_O_3_	SiO_2_	CaO	MgO	Na_2_O	MnO	Other
1	30.4	19.5	33.8	6.5	1.6	0.8	1.3	6.1
2	58.0	8.4	21.7	3.4	1.5	0.6	1.0	5.4

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
