# Peer review of "Coal Ash Enrichment with Its Full Use in Various Areas"

_materials, 2022, doi:10.3390/ma15196610_

Round 1

Reviewer 1 Report

This is an interesting and well elaborated paper aimed at the coal ash enrichment and its fully application in various industrial areas. From my point of view, the manuscript does not need any substantially improvements, and it can be accepted for publication as following comment and suggestions will be considered in its minor revision. See my particular comments below.

i)                 In Fig. 5, it is not clear whether the composition of PC is presented in wt.% or volume %.

ii)                How was the dosage of plasticizer chosen? Did the authors some rheology tests?

iii)               I don’t understand why compressive strength testing was done on 70 mm cubes. Usually 100 mm cubes or 160/40/40 mm prisms are used in strength testing.

iv)               Fig. 6 – the cement replacement was calculated as mass ratio? The dosage of cement is introduced in kg? Similarly water and plasticizer? Make it clear.

v)                What was the uncertainty of the compressive strength test?

vi)               In Fig. 19, error bars should be presented.

vii)             That’s problem, the compressive strength for was the only measured parameter of the hardened samples. Some information on materials density, specific density, and porosity might be given.

Author Response

Dear Reviewer 1!

Thank you for your interest in our manuscript. Your valuable comments helped make the manuscript even better. All corrections in the manuscript are highlighted in blue.

Comment 1: In Fig. 5, it is not clear whether the composition of PC is presented in wt.% or volume %.

Response: Added (wt. %)

Comment 2: How was the dosage of plasticizer chosen? Did the authors some rheology tests?

Response: Dosages of plasticizer seen in Fig. 6

Comment 3: I don’t understand why compressive strength testing was done on 70 mm cubes. Usually 100 mm cubes or 160/40/40 mm prisms are used in strength testing.

Response: The compressive strength was determined according to the Russian standard GOST 310.4-81

Comment 4: Fig. 6 – the cement replacement was calculated as mass ratio? The dosage of cement is introduced in kg? Similarly water and plasticizer? Make it clear?

Response: Added (in fractions)

Comment 5: What was the uncertainty of the compressive strength test?

Response: Added: «The scatter of the results obtained for all tests within each series of samples did not exceed 5%»

Comment 6:  In Fig. 19, error bars should be presented.

Response: The scatter of the results obtained for all tests within each series of samples did not exceed 5%

Comment 7: That’s problem, the compressive strength for was the only measured parameter of the hardened samples. Some information on materials density, specific density, and porosity might be given.

Response: Due to the fact that the article is devoted to the issues of complex utilization of coal ash not only for building applications, only the most important and indicative tests were included.

Reviewer 2 Report

Please, see the comments in the attachment.

Author Response

Dear Reviewer 2!

Thank you for your interest in our manuscript. Your valuable comments helped make the manuscript even better. All corrections in the manuscript are highlighted in blue.

Comment 1: Page 3: “Researchers often use milling to increase the activity of fly ash [41]”. Observation: Is this a sustainable approach? Expecially taking into account the current energetic problems, each of these operations adds significant energy costs. Authors should (preliminarly) evaluate whether their choices are feasible and economically sustainable not only from a research perspective, but also from an industrialapplicative one. From this perspective, similar assessments must be made for the other proposed operations, thus justifying the real effectiveness of the proposed technology.

Response: Added: «However, ash milling technology should be treated with caution as it is not always a sustainable approach. Especially given the current energy challenges, each of these operations adds significant energy costs. When designing compositions, researchers should (preliminarily) evaluate the feasibility and economic feasibility of their choice, not only from a research point of view, but also from an industrial and applied point of view. From this point of view, similar assessments should be made for other proposed operations, thereby confirming the real effectiveness of the proposed technology. On the other hand, without grinding or other enrichment, it is impossible to turn waste into secondary resources.»

Comment 2: Page 3: “Such ash is characterized by a heterogeneous composition, low calcium content, and the presence of a sufficiently large volume of impurities [51]”. Observation: The Authors bring out the issue of compositional variability of the ashes. What effect could this have on the effectiveness of the proposed treatments? Would this technology be robust and flexible? The Authors should adequately comment on this issue.

Response: Added: «The issue of the ash composition variability is raised before researchers, due to the fact that this can affect the effectiveness of mechanical processing methods. For enrichment technology to be reliable and flexible, it must be complex and involve more than just grinding.»

Comment 3: Page 3: “Causes difficulties in the processing of acidic fly ash in the production of modern cements and concretes, the variable composition of particles in size [52]. First of all, this concerns the content of particles in the range of the largest fractions in the composition of the ash and slag mixture [53]. Therefore, the ash waste from hydraulic removal must be separated into separate fractions. Thus, it will be possible to use each individual component exactly where it will make the greatest contribution to the creation of building products [54]. Separation, flotation, dispersion, activation in various grinding devices and other methods of ash processing contribute to its efficiency [55]”. Observation: The Authors focus a lot on the fractionation of the coal ash components, direct and immediate applications of the bulk ashes could be possible for the production of new materials, for example in the road construction (e.g. formulation of asphalts). Direct use of wastes should be preferred, avoiding additional costs, expecially for final materials of low-value. Again, the Authors should adequately comment and justify their choices..

Response: Added: «Of course, direct and immediate use of bulk ash is possible for the production of various building materials, for example in road construction (eg asphalt composition). It is more economical to prioritize the direct use of waste, avoiding additional costs, especially for low value end materials. However, given the potential activity of enriched coal ash, it is unreasonable to "bury" this valuable resource, for example, as bedding under road pavement. Moreover, the huge areas of ash dumps force us to look for and use all possible methods of disposal.»

Comment 4: Page 4: “Ash from the Kashirskaya thermal power plant (Russia) was chosen as the object of study (Fig. 2)”. Observation: If possible, Authors should (approximately) include the amounts produced on an annual basis, to set up a material balance, which includes also the recovered fractions, and possibly make some cost considerations. Moreover, Authors should give details on ash sampling, to ensure the representativeness of the samples (useful for evalutation of particle size distribution). In this sense, the Authors claim at page 7: “Mathematical planning was carried out using the Develve software package. The representativeness of the sample of the number of samples and the set of necessary tests was carried out from modern positions; in this case, the error for all experiments was no more than 5%.”. In my opinion, Authors should further deepen not only the mathematical processing, but also the sampling activity.

Response: Added: «The annual amount of coal ash emitted from this thermal power plant is 9 million tons.» Also added Fig 6 that gives details on ash sampling, to ensure the representativeness of the samples

Comment 5: Page 6: “The effectiveness of the obtained aluminosilicate component of enriched coal ash was tested during its use as a replacement for cement in various proportions (Fig. 6).”. Observation: Authors analyze defined compositional ranges of the components of interest (chosen on the basis of what criteria?): essentially data reported in Figure 6 define a matrix. To improve the quality of the paper, the Authors could analyze their data with a multivariate analysis software, to evaluate the effect of the proposed range of composition of the components on the responses of their interest.

Response: The composition ranges of the components of interest are selected based on preliminary studies and multivariate analysis software. Due to the fact that the article is devoted not only to the study of these compositions for building use, but to the complex utilization of coal ash, the results of a preliminary selection of building compositions are not given.

Comment 6: Page 8: “The coarse fraction is sent to the dump, and the fine particles enter the flotation unit, where the underburnt is removed (Fig. 8)… The resulting underburnt is briquetted and used as a fuel (Fig. 9).”. Observation: Detailed characterization data of the unburned coal component should be added and properly discussed, to demonstrate the effectiveness of the flotation unit and the use of this fraction as a fuel. On page 14, the Authors report some preliminary characterization data “Density of obtained fuel briquettes is 1000-1200 kg/m3, calorific value is 19.5-20 MJ/kg, ash content is 0.5-1.5%.”, 73 / 5.000”, but it is necessary to pursue a systematic characterization of the unburned coal fraction (at least ultimate and proximate analysis data), as well as the chemical composition of the ashes (to evaluate for improving data discussion of the other fractions).

Response: Added Table 2 that lists the chemical composition of the underburnt.

Comment 7:  Page 9: “At the same time, at the first stage, the separator acts on the coal ash with a magnetic induction of 600 T, removing various contaminants together with iron-containing particles; at the next stage, a more precise cleaning with a magnetic induction of 400 T takes place.”. Observation: Again, the Authors should give specific and detailed information on the characterization of these two fractions, to prove their claims. In this way, it becomes clearer why a two-step treatment is necessary to improve the properties of the final accepted fraction.

Response: Added Table 3 that lists the chemical composition of the iron-containing component after each magnetic separation stage

Comment 8:  Page 11: “Grinding was carried out in an Activator-4M planetary mill (Activator, Chelyabinsk, Russia) for 30 minutes.”. Observation: This sentence must be moved to the "Materials and Methods" Section.

Response: Moved

Comment 9: Page 11: “The impact of ferromagnetic grinding media (Fig. 16a) allows efficient milling and activation of the powder (Fig. 16b).”. Observation: Again, the treatment must be sustainable and economically feasibile, to be of real industrial and practical interest. Could other choices be made to improve the sustainability of this step?

Response: Added: «Processing must be sustainable and economically viable to be of real industrial and practical interest. Against the background of other options for increasing the stability of this step, the use of a planetary mill allows, in the shortest possible time and at minimal cost, not only to grind, but also to activate the powder due to the combined action of impact, centrifugal and abrasive forces.»

Comment 10:  Page 14: “The resulting fuel briquettes do not include any binders, except for one natural such lignin, contained in the cells of plant waste.”. Observation: What kind and how much lignin has been used as the binder?

Response: In this sentence, it has been added that hydrolytic lignin was found in an amount of 7 wt. %

Comment 11:  Page 14: “The iron-bearing component recovered by two-stage magnetic separation has the potential to be used in metallurgy as a coking additive, in particular for the production of iron and steel.”. Observation: Why is it not possible to use the iron-bearing fraction resulting from a single magnetic separation stage? Authors should prove their claims, especially if this component should be used for different applications. For the metallurgical applications, a two-stage magnetic separation treatment will be necessary (I suppose), because the quality of the final product must be high and well-defined, therefore without including unwanted impurities. On the other hand, a two-stage magnetic separation is probably unnecessary, and therefore economically disadvantageous for developing low-cost agronomic applications. As previously stated in one comment, the characterization data of the two fractions could help the Authors to support their conclusions.

Response: Added: «In fact, it is possible to use the iron fraction obtained from a single magnetic separation step to develop low-cost agronomic applications. However, for metallurgical applications a two-stage magnetic separation will be necessary because the quality of the final product must be high and well defined, therefore without the inclusion of undesirable impurities. The chemical composition of the iron-containing component obtained after each stage, shown in Table 3, will help readers determine the required application of the materials.»

Comment 12: “In metallurgy, the use of this additive in metal smelting has a key role”. Observation: Please, add appropriate bibliographic references that justify this sentence, as well as the related subsequent ones.

Then, proceed similarly for the agronomic part, e.g. “The obtained iron-containing component is also an effective microfertilizer” and subsequent corresponding sentences. Furthermore, these applications are inferred only on a literature basis, whilst the Authors could give experimental details about agronomic and metallurgical uses, if available for this specific and real case of study.

Response: Added: «Not only iron-containing components are used in metallurgy, but production waste is also used in the aluminum industry [68].

This confirms by study [69] that provides a green engineering approach to recycle coal ash for regreening mines, as well as a new development direction for high-value green recyclable pathway of coal ash.

In fact, it is possible to use the iron fraction obtained from a single magnetic separation step to develop low-cost agronomic applications. However, for metallurgical applications a two-stage magnetic separation will be necessary because the quality of the final product must be high and well defined, therefore without the inclusion of undesirable impurities. The chemical composition of the iron-containing component obtained after each stage, shown in Table 3, will help readers determine the required application of the materials.»

Comment 13:  Page 15: “Conclusions” Section. Observation: The Conclusions should be improved, taking into account the comments up to now proposed to the Authors.

Response: Conclusions have been improved

Comment 14:  Other observations: Please, improve the quality of Figures related to particles size distributions (the resolution is rather poor).

Response: Improved

Reviewer 3 Report

A significant step in protecting the environment is raising the amount of industrial waste that is recycled. One of the largest tonnage waste products produced by the operation of thermal power plants is coal ash.

1.      I suggest updating the abstract to include the most recent information. It is also necessary to include the problem statement. According to the abstract, the outcome must have been actual.

2.      As a second suggestion, I'd propose including an instance of the paper's structure at the end of the introduction section. This manuscript's remaining sections are structured as follows. The following is found in Section 2: In Section 3......... in Section 4, etc.

3.      For the convenience of all readers, the authors included a Nomenclature List.

4.      Fig. 3. Chemical composition of the original coal ash’’ should be clear  

5.      Ash from the Kashirskaya thermal power plant (Russia) was chosen as the object of study (Fig. 2). Page 4, section 2, should be clear and rewritten.

6.      The particle size distribution is shown in Fig. 4. Page 4, section 2, should be clear and rewritten.

7.       Fig. 10. Magnetic separation should be clear in the caption name and illustration.

8.      Please double-check the references update.

9.      The conclusion should have more specific numbers No. 5.

Author Response

Dear Reviewer 3!

Thank you for your interest in our manuscript. Your valuable comments helped make the manuscript even better. All corrections in the manuscript are highlighted in blue.

Comment 1: I suggest updating the abstract to include the most recent information. It is also necessary to include the problem statement. According to the abstract, the outcome must have been actual.

Response: Abstract has been updated. Also added: «problem statement lies in the fact that complex and energy-consuming technologies of mechanical activation and enrichment minimize the effect of the use of coal ash»

Comment 2:  As a second suggestion, I'd propose including an instance of the paper's structure at the end of the introduction section. This manuscript's remaining sections are structured as follows. The following is found in Section 2: In Section 3......... in Section 4, etc.

Response: Added: «This manuscript's remaining sections are structured as follows. Section 2 detailed materials and methods. In Section 3.1 a technology of coal ash complex enrichment and separation to components is shown. Section 3.2 is about use of the aluminosilicate part as a pozzolanic additive to cement. Section 3.3 studied carbon underburning for fuel briquettes. Section 3.4 detailed iron-containing part for metallurgy and agriculture. Section 4 summarized some conclusions for conducted research.»

Comment 3: For the convenience of all readers, the authors included a Nomenclature List.

Response: Nomenclature List has been added to the end of the paper

Comment 4: Fig. 3. Chemical composition of the original coal ash’’ should be clear 

Response: Cleared

Comment 5: Ash from the Kashirskaya thermal power plant (Russia) was chosen as the object of study (Fig. 2). Page 4, section 2, should be clear and rewritten.

Response: Corrected: « Object of the study is the coal ash from the Kashirskaya thermal power plant (Kashira, Russia) »

Comment 6:  The particle size distribution is shown in Fig. 4. Page 4, section 2, should be clear and rewritten.

Response: Cleared and rewritten

Comment 7: Fig. 10. Magnetic separation should be clear in the caption name and illustration.

Response: Corrected: «Fig. 12. Magnetic separator (a) and obtained material (b).»

Comment 8:  Please double-check the references update.

Response: Double-checked

Comment 9: The conclusion should have more specific numbers No. 5.

Response: Conclusions have been improved

Round 2

Reviewer 2 Report

The authors have made the proposed changes and the manuscript can be accepted. Especially for their future work on such real case studies, I suggest they should include detailed economic evaluations of the involved unit operations, to add more value to their work. Otherwise, it is not possible to understand if their work can be really useful for the valorization of these waste fractions and their proposals remain limited to the academic scale.